# Gene Expression Linked to Reepithelialization of Human Skin Wounds

**DOI:** 10.3390/ijms232415746

**Published:** 2022-12-12

**Authors:** Magnus S. Ågren, Thomas Litman, Jens Ole Eriksen, Peter Schjerling, Michael Bzorek, Lise Mette Rahbek Gjerdrum

**Affiliations:** 1Department of Dermatology and Copenhagen Wound Healing Center, Bispebjerg Hospital, University of Copenhagen, 2400 Copenhagen, Denmark; 2Digestive Disease Center, Bispebjerg Hospital, University of Copenhagen, 2400 Copenhagen, Denmark; 3Department of Clinical Medicine, Faculty of Health and Medical Sciences, University of Copenhagen, 2200 Copenhagen, Denmark; 4Department of Immunology and Microbiology, Faculty of Health and Medical Sciences, University of Copenhagen, 2200 Copenhagen, Denmark; 5Department of Pathology, Zealand University Hospital, 4000 Roskilde, Denmark; 6Institute of Sports Medicine Copenhagen, Department of Orthopedic Surgery, Copenhagen University Hospital—Bispebjerg-Frederiksberg, 2400 Copenhagen, Denmark; 7Center for Healthy Aging, Department of Clinical Medicine, Faculty of Health and Medical Sciences, University of Copenhagen, 2200 Copenhagen, Denmark

**Keywords:** wound healing, gene expression, keratinocytes, fibroblasts, cytokines, matrix metalloproteinases

## Abstract

Our understanding of the regulatory processes of reepithelialization during wound healing is incomplete. In an attempt to map the genes involved in epidermal regeneration and differentiation, we measured gene expression in formalin-fixed, paraffin-embedded standardized epidermal wounds induced by the suction-blister technique with associated nonwounded skin using NanoString technology. The transcripts of 139 selected genes involved in clotting, immune response to tissue injury, signaling pathways, cell adhesion and proliferation, extracellular matrix remodeling, zinc transport and keratinocyte differentiation were evaluated. We identified 22 upregulated differentially expressed genes (DEGs) in descending order of fold change (*MMP1*, *MMP3*, *IL6*, *CXCL8*, *SERPINE1*, *IL1B*, *PTGS2*, *HBEGF*, *CXCL5*, *CXCL2*, *TIMP1*, *CYR61*, *CXCL1*, *MMP12*, *MMP9*, *HGF*, *CTGF*, *ITGB3*, *MT2A*, *FGF7*, *COL4A1* and *PLAUR*). The expression of the most upregulated gene, *MMP1*, correlated strongly with *MMP3* followed by *IL6* and *IL1B*. rhIL-1β, but not rhIL-6, exposure of cultured normal human epidermal keratinocytes and normal human dermal fibroblasts increased both *MMP1* mRNA and MMP-1 protein levels, as well as *TIMP1* mRNA levels. The increased *TIMP1* in wounds was validated by immunohistochemistry. The six downregulated DEGs (*COL7A1*, *MMP28*, *SLC39A2*, *FLG1*, *KRT10* and *FLG2*) were associated with epidermal maturation. *KLK8* showed the strongest correlation with *MKI67* mRNA levels and is a potential biomarker for keratinocyte proliferation. The observed gene expression changes correlate well with the current knowledge of physiological reepithelialization. Thus, the gene expression panel described in this paper could be used in patients with impaired healing to identify possible therapeutic targets.

## 1. Introduction

Reepithelialization during wound healing is crucial for restoring the skin barrier. Our understanding of the regulatory processes is incomplete for this fundamental process. The factors responsible have not been fully delineated in humans [1,2], although there are gene expression data from partial-thickness excisional and burn wounds [3,4] as well as full-thickness skin wounds in patients with basal cell carcinoma [5].

The suction-blister wound healing model is excellent for studies of the reepithelialization process, and this model has been used to evaluate systemic and topical factors and interventions [1,6,7,8,9,10,11,12,13,14,15,16,17]. For example, treatment with a general matrix metalloproteinase (MMP) inhibitor has been shown to delay reepithelialization [11]. Many clinical outcomes have been validated, but more knowledge of the underlying molecular mechanisms is needed.

The nCounter^®^ (NanoString Technologies, Seattle, WA, USA) gene expression assay is based on direct digital detection of mRNA molecules using target-specific, color-coded probe pairs. It does not require the conversion of mRNA into cDNA by reverse transcription or the amplification of the resulting cDNA by PCR, limiting analytical bias. Another important feature is that the technology can be applied directly to RNA extracted from formalin-fixed, paraffin-embedded (FFPE) tissues [18].

The aim of this study was to validate a customized gene expression panel using archival FFPE tissues of epidermal wounds induced by the suction blister technique from a randomized, double-blind controlled trial in healthy volunteers [1,14,15].

## 2. Results

To screen candidate genes involved in normal epidermal wound healing, we designed a gene expression panel composed of 139 different target genes encoding transcription factors, cytokines/chemokines, growth factors, receptors, extracellular matrix (ECM) molecules, proteinases/antiproteinases, zinc importers/exporters, antibacterial peptides, adhesion molecules and epidermal stratification markers (Appendix B, Table A1).

We analyzed FFPE tissues from 4-day-old epidermal wounds (reepithelialized to 30–40% as determined by histology [14]), including adjacent normal skin from 20 nondiabetic participants (age 19–43 years old, 27.2 ± 6.0 years). The 8 women and 12 men were included in the period from 30 March 2014 to 4 May 2014 [1,14,15]. Six participants had skin type I, three had type II, seven had type III and four had skin type IV [19].

### 2.1. Determination of mRNAs by NanoString

The raw barcode counts of the immobilized labeled mRNA complexes were obtained by automated scanning using inverted fluorescence microscopy. The data for all samples and genes are shown in Appendix A. The expression of the genes was then normalized to the spiked-in positive controls and six different housekeeping genes using nSolver software. The housekeeping genes were expressed at similar levels for all samples. The normalized values are shown in Appendix A and were used for the subsequent analyses. 

Skin samples of three participants (24, 27 and 28) were flagged according to the nSolver default algorithm due to low RNA contents; therefore, paired analyses with the wounds were not possible. The comparisons between wounds and skin, expressed as fold changes (FC) with *p* and *q* values, for the remaining 17 participants are shown for the 139 genes in Appendix A.

### 2.2. Differentially Expressed Genes (DEGs) in Human Epidermal Wounds

We defined a DEG as a gene with a mean 2-FC and *p* < 0.05 in the wound group compared to the associated nonwounded skin. When using this criterion, the following 28 DEGs (all *p* < 0.05, *q* < 0.05) arranged in descending order of FC were: *MMP1*, *MMP3*, *IL6*, *CXCL8*, *SERPINE1*, *IL1B*, *PTGS2*, *HBEGF*, *CXCL5*, *CXCL2*, *TIMP1*, *CYR61*, *CXCL1*, *MMP12*, *MMP9*, *HGF*, *CTGF*, *ITGB3*, *MT2A*, *FGF7*, *COL4A1* and *PLAUR* (22 upregulated DEGs) and *COL7A1*, *MMP28*, *SLC39A2*, *FLG1*, *KRT10* and *FLG2* (6 downregulated DEGs), as illustrated in Figure 1. 

### 2.3. MMP1 mRNA Correlations in Wounds

*MMP1* was the most upregulated gene in the wounds compared to the skin. A first step to elucidate the possible regulatory mechanisms of *MMP1* was to correlate *MMP1* mRNA with the expression of the other upregulated genes. The strongest correlation was found for *MMP3* (*r* = 0.83, *p* = 4.9 × 10^−6^) followed by the expression of *IL6* (*r* = 0.70, *p* < 0.001) and *IL1B* (*r* = 0.59, *p* < 0.01), as shown in Figure 2. *IL1B* mRNA levels correlated strongly (*r* = 0.69, *p* < 0.001) with *IL6* mRNA levels in wounds (Appendix A).

### 2.4. Effects of Cytokines on MMP1, MMP3 and TIMP1 Gene Expression and Secretion of MMP-1 and TIMP-1 into the Medium of Normal Human Epidermal Keratinocytes (NHEKs) and Normal Human Dermal Fibroblasts (NHDFs)

To study the transcriptional regulation of *MMP1, MMP3* and *TIMP1* and translation into MMP-1 and TIMP-1, NHEKs and NHDFs were exposed to 30 ng/mL rhIL-6 and 1 ng/mL rhIL-1β separately and together. The IL-6 and IL-1β concentrations were chosen from measurements of 1-day-old suction blister wounds [20,21] and were found to be noncytotoxic, as indicated by the similar LDH activities in the media (Table 1).

Previous studies in NHDFs indicated that IL-6 is anti-proliferative at ≥10 ng/mL while IL-1β (0.1 ng/mL) is proliferative [22]. In our studies of NHDFs using the BrdU incorporation assay [23,24], IL-6 and IL-1β did not significantly influence DNA synthesis at 30 ng/mL and 1 ng/mL, while IL-6 at 100 ng/mL tended to reduce (*p* = 0.052) BrdU incorporation into NHDFs (Appendix A).

#### 2.4.1. MMP1 and TIMP1 mRNA and MMP-1 and TIMP-1 Protein Levels

IL-6 treatment for 24 h increased *MMP1* mRNA levels 1.6-fold in NHEKs and 2.5-fold in NHDFs. The corresponding increases for IL-1β were 2.0-fold and 85-fold. Utani et al. [25] reported a 2.5-fold increase in *MMP1* mRNA in NHEKs and a 63-fold increase in NHDFs after 8 h of IL-1β (10 ng/mL) treatment. There were no additive effects of combining IL-6 with IL-1β treatment on the *MMP1* mRNA levels in NHEKs (*p* = 0.641) or in NHDFs (*p* = 0.362). MMP-1 protein levels in control conditioned media of NHEKs were 46.5 ± 6.6 ng/mL and of NHDFs 322 ± 53 ng/mL. IL-6 had no significant effect on MMP-1 protein levels while IL-1β increased MMP-1 protein levels in NHEKs (1.7-fold) and NHDFs (3.3-fold) compared to the controls. No additive effect of IL-6 was observed in either cell type (Figure 3).

*TIMP1* mRNA levels were increased in NHEKs (1.2-fold) and in NHDFs (3.6-fold) with IL-1β, but not with IL-6 exposure. The TIMP-1 protein concentration of the control medium of NHEKs was 19.1 ± 3.1 ng/mL and that of the NHDFs was 170 ± 23 ng/mL. IL-1β increased TIMP-1 protein levels in conditioned media of NHDFs (1.2-fold) but not of NHEKs. IL-6 treatment was ineffective (Figure 3).

#### 2.4.2. MMP3 mRNA Levels

In NHEKs, only the combination of IL-6 and IL-1β upregulated *MMP3* mRNA levels. IL-6 had no significant effect on *MMP3* mRNA levels in NHDFs, in contrast to the 121-fold stimulation (*p* < 0.001) by IL-1β of *MMP3* mRNA in NHDFs (Appendix A).

### 2.5. TIMP-1 and Collagen I Protein Expression

We applied immunohistochemistry to validate the increased *TIMP1* mRNA levels in wounds at the protein level and to elucidate the cellular sources of TIMP-1 in 19 wounds with adjoining skin. TIMP-1 was undetectable in the epidermal compartments of skin and wounds. In the dermal compartments, a few fibroblasts revealed TIMP-1 staining in normal skin, while the expression of the TIMP-1 protein was markedly increased in fibroblasts and in endothelial cells, as well as lymphocytes in wounds, compared to normal skin. TIMP-1 was observed in hair follicle epithelium more often in wounds than in the skin. Acrosyringium of the eccrine sweat gland was positive in one wound. In general, TIMP-1 stained granularly in the cytoplasm of the cells (Figure 4A,B). 

Collagen I strongly stained the dermis in both wounds and skin and was observed below the basement membrane but not in epidermis (Figure 4C,D).

### 2.6. Proliferation Markers in Wounds

To delineate conceivable biomarkers of cell proliferation, we examined correlations between *MKI67* mRNA levels and the entire panel of genes. *KLK8* (*r* = 0.90, *p* = 7.2 × 10^−8^) and *KRT6A* (*r* = 0.83, *p* = 5.8 × 10^−6^) showed strong correlations with *MKI67* mRNA levels, as shown in Figure 5.

## 3. Discussion

The aim of this study was to design and validate an expression panel of genes involved in reepithelialization during wound healing. Unlike high-throughput next-generation technologies, such as microarray and RNASeq, which produce a comprehensive set of gene expression profiles, we selected genes known to be involved in wound healing from the literature. This focused strategy lowers the false discovery rate (FDR) with accompanying increased power. Of the 22 upregulated DEGs, nine were related to ECM remodeling, six encoded cytokines/chemokines and five were growth factor-transcribing genes, and all of these genes are considered important for reepithelialization.

*MMP1* was the most upregulated gene in the wounds. Nuutila et al. [3] reported that *MMP1* was the top overexpressed gene in split-thickness skin graft (STSG) donor site wounds using genome-wide transcriptomic methodology. MMP-1 is unambiguously associated with reepithelialization [26] and we recently demonstrated the exclusive presence of the MMP-1 protein in the neoepidermis and in fibroblasts in the dermis beneath the neoepidermis [1].

The regulation of the *MMP1* gene involves several signaling pathways [1,27,28]. The strong correlation between *IL6* and *MMP1* mRNA levels indicated a possible role of IL-6 in the induction of *MMP1*. The effect of IL-6 treatment of cultured NHEKs and NHDFs was weak in contrast to IL-1β. These somewhat contradictory results could be explained by the indirect effect of IL-1β on *IL6* expression [29]. We also found that *IL1B* mRNA levels correlated strongly with *IL6* mRNA levels in the wounds.

It has been suggested that collagen I is the primary inducer of *MMP1* via the α2β1 integrin [30]. Collagen I was observed to be juxtaposed to the basement membrane by immunohistochemistry, implying no direct contact of keratinocytes with collagen I, indicating that collagen I-keratinocyte interactions are subordinate in the regulation of *MMP1*. It should be emphasized that the basement membrane remains essentially intact during reepithelialization [1,28]. The elevated *COL4A1* may indicate the requirement for collagen IV by migrating keratinocytes on the basement membrane but also for angiogenesis [28,31].

*MMP1* belongs to the MMP family, consisting of 23 human members [2,32]. In another study, *MMP1* and *MMP3* expression was coordinately induced [33]; the *MMP1* and *MMP3* genes are closely located on chromosome 11q22.2. *MMP3* was also the second most upregulated gene and was increased by IL-1β but not by IL-6. In earlier studies, investigators failed to detect *MMP3* in suction blister wounds by in situ hybridization [11,28]. This discrepancy might be attributed to the high sensitivity of our analyses. MMP-3 contributes to the conversion of latent into active MMP-1 [34]. Apart from *MMP1* and *MMP3*, *MMP12* and *MMP9* were upregulated. MMP-12 appears to play a key role in cytoskeletal rearrangements in migrating keratinocytes [35]. MMP-9 protein increases during suction blister wound healing [7,11].

MMP activity is antagonized by tissue inhibitors of metalloproteinases (TIMPs), and *TIMP1* transcripts were increased in wounds compared with adjacent skin. Our immunohistochemical analysis clearly demonstrated increased TIMP-1 protein expression in wounds vs. skin primarily due to TIMP-1-producing fibroblasts, endothelial cells, and lymphocytes. In one study, *TIMP1* mRNA was expressed in fibroblasts below the neo-epidermis but absent in keratinocytes [28]. Mechanistically, inhibition of the proteolytic action of MMP-1 slows epidermal tongue movement. Local TIMP-1 overexpression retards keratinocyte migration in wounds [36]. It has also been suggested that TIMP-1 is a beneficial angiogenic factor [37].

Regulation of *TIMP1* has rarely been described. IL-6 did not induce the *TIMP1* gene in NHDFs, corroborating earlier findings [38]. We found that IL-1β induced *TIMP1*, which also resulted in increased TIMP-1 secretion from NHDFs [39].

Neither *COL1A1* expression nor proliferation of stromal cells were increased in dermis [1]. The lack of a fibroproliferative response explains the fact that these lesions leave no scar.

The precursor plasminogen of the serine proteinase plasmin acts together with MMPs for maximal keratinocyte migration [40,41]. Plasminogen activator inhibitor *SERPINE1* and the receptor for urokinase plasminogen activator *PLAUR* were upregulated, indicating the need for the plasminogen system in coordinating reepithelialization [42,43,44].

Different soluble mediators are involved in the immunoregulatory and inflammatory processes in these lesions [20]. The protein levels of IL-6, IL-8 and IL-1β were dramatically increased 1 day after wound infliction while IL-1α levels remained low [20]. These data are convincingly consistent with our mRNA measurements of *IL6*, *CXCL8* and *IL1B*. In contrast, *CSF2*, *IFNG*, *IL1A* and *TNF* were not upregulated in the wounds vs. nonwounded skin. This finding does not exclude the possibility that these cytokines were upregulated earlier. The proinflammatory chemokines *CXCL5*, *CXCL2* and *CXCL1* in addition to *CXCL8* were upregulated in wounds, presumably as a response to injury [45]. The main receptor CXCR2 is expressed on innate immune cells and on keratinocytes; CXCR2-deficient keratinocytes display impaired migration [46]. Prostaglandin-endoperoxide synthase 2 (*PTGS2*), also known as COX-2, is induced by trauma in the epidermis [47] and provides proinflammatory prostaglandins.

Five growth factors (*HBEGF*, *CYR61*/*CCN1*, *HGF*, *CTGF*/*CCN2* and *FGF7*) were upregulated in the wounds as indicated in previous studies [48,49,50,51,52]. The neoepidermis is regenerated by the combined action of keratinocyte migration and proliferation. Previously, we showed that keratinocyte proliferation was increased to the same magnitude in neoepidermis and adjacent epidermis [1]. This finding might explain the lack of difference in gene expression of the proliferation marker *MKI67* between wounds and skin. Collectively, these findings imply that the primary mode of action of the upregulated growth factors is the stimulation of keratinocyte migration independent of their mitogenic effects; these factors also have the ability to enhance the migration of epidermal keratinocytes in vitro [48,49,50,51,53].

The serine proteinase kallikrein-related peptidase 8 (*KLK8*) showed the strongest correlation with *MKI67*. KLK8 is one of 15 different kallikreins in the epidermis involved in epidermal homeostasis [54]. Interestingly, delayed wound healing in *KLK8*-knockout mice was accompanied by decreased Ki-67 immunolabeling of the neoepidermis [55]. Keratin-6A (*KRT6A*) mRNA levels also correlated strongly with *MKI67* gene expression. Keratin-6A protein levels were increased in proliferating vs. differentiating NHEKs in vitro [56], which was the reason why we investigated the usefulness of keratin-6A protein as a biomarker for keratinocyte proliferation in human wounds in a previous study [56]. However, keratin-6A was undetectable possibly because it is an intracellular protein [56].

The increased *MT2A* mRNA levels in wounds corroborate our previous immunohistochemical results using an anti-MT antibody that reacted not only with MT2A but also with the subisoform MT1A [1]. Because MT1A was not significantly upregulated in the wounds compared to skin, MT2A was most likely the predominant metallothionein isoform detected in the wounds [1].

The downregulated genes were associated with epidermal maturation. Collagen VII is the main component of anchoring fibrils. Nyström et al. [57] observed collagen VII in migrating epidermis, but this finding was in wounds devoid of basement membrane. Leivo et al. [10] found no differences in collagen VII between suction blister floor and normal skin. The role of MMP-28 is unclear; *MMP28* mRNA was detected distal to the leading edge in the proliferating epidermal compartment [58]. The zinc importer *SLC39A2* (ZIP2) has been suggested to participate in keratinocyte differentiation [59]. *K10* is an early epidermal differentiation marker. Filaggrins (*FLG1* and *FLG2*) are essential for the formation of a functional stratum corneum [60,61].

## 4. Materials and Methods

### 4.1. Ethical Statements

The study was approved by the Committee on Biomedical Research Ethics for the Capital Region of Denmark (H-6-2014-001) and was registered at ClinicalTrials.gov (NCT02116725) on 15 April 2014, and conducted at the Department of Dermatology, Bispebjerg Hospital, University of Copenhagen, Copenhagen, Denmark [14].

### 4.2. Participants

Healthy nonsmoking volunteers between 18 and 65 years of age were included after providing written informed consent. Individuals with skin disorders; those who were pregnant, breastfeeding or receiving systemic immunosuppressive treatment; and/or those who were hypersensitive to zinc were excluded [1,14,15].

### 4.3. Induction of Epidermal (Suction Blister) Wounds, Treatment and Tissue Procedures

Suction blisters (10 mm in diameter) were raised on each buttock in 30 remunerated participants, and the blister roofs were excised. Two of the three treatments (zinc sulfate, placebo, or control) were randomized to the left or right wound by concealed allocation, i.e., each of the 3 treatments was applied to 20 wounds in 20 participants. In the present study, the participants of the control arm of this three-arm randomized, double-blind trial were included [14]. Wounds and adjoining skin were treated once daily with distilled water and covered with a bacteria-proof and moisture-retaining dressing (Mepore Film & Pad, Mölnlycke Health Care, Göteborg, Sweden) [1,15]. On post-wounding day 4, wounds, including uninjured skin, were excised [1]. The biopsies were fixed in 4% phosphate-buffered paraformaldehyde (pH 7.4) overnight at 4 °C and embedded in paraffin.

### 4.4. Macrodissection and Isolation of RNA from FFPE Wound and Skin Compartments

Tissue sections were cut at 5 µm and dissected into one central wound piece and two adjacent pieces of normal nonwounded skin (Figure 6).

The tissue pieces were deparaffinized in xylene and absolute ethanol, and the tissue was scratched into reaction vials using a scalpel. Total RNA was extracted with a High Pure FFPE RNA Micro Kit (Roche, Mannheim, Germany). The concentration and purity of RNA in 20 µL elution buffer were determined by NanoDrop (NanoDrop Technologies, Wilmington, DE, USA) spectrophotometry. RNA purity was indicated by the OD_260 nm_/OD_280 nm_ ratio (≥1.5 was acceptable).

### 4.5. Design of Gene Panel and nCounter Analyses

The NanoString human Preselected PlexSet Wound Healing Panel was used encompassing 90 target genes related to clotting, immune response to tissue injury, and ECM remodeling, along with relevant signaling pathway genes coordinating wound healing. Forty-nine wound-healing related genes expressed in skin were added to this panel by the authors, as indicated in Appendix B (Table A1). The details of the design of our gene expression panel are found in Appendix A.

The isolated total RNA (100 ng) was hybridized to the code sets at 65 °C overnight on the nCounter Prep Station. Gene expression was analyzed with an nCounter Digital Analyzer. Fluorescence was determined using a built-in inverted fluorescence microscope using all 550 possible counting areas of the NanoString cartridges.

nSolver software, version 4.0, was used to export, quality check, and normalize the hybridization results. Background subtraction was performed by negative control thresholding using the average of the included 8 probe-set negative controls. Normalization was performed in two steps. First, based on the geometric mean of the spiked-in positive controls, the positive control normalization factor was calculated to adjust for differences in various steps of the process. Default settings between 0.3 and 3 were used. Second, to adjust for differences in analyte abundance and quality across samples, a normalization factor was calculated based on the geometric mean of the included housekeeping genes (*ABCF1*, *GUSB*, *HPRT1*, *LDHA*, *PTEN* and *RPLP0*). The default settings (0.1–10) were applied.

### 4.6. Immunohistochemical Analysis of TIMP-1 and Collagen I

Sections with a thickness of 4 μm were cut, and slides were deparaffinized and rehydrated. The sections were pretreated and stained using the Omnis automated slide-processing system from Agilent (Glostrup, Denmark). The tissue sections were subjected to heat-induced epitope retrieval pretreatment using EnVision™ FLEX Target Retrieval Solution High pH (GV804, Dako Omnis, Agilent) for TIMP-1 or EnVision™ FLEX Target Retrieval Solution Low pH (GV805, Dako Omnis, Agilent) for collagen I for 30 min, followed by incubation with rabbit monoclonal antibodies against TIMP-1 (clone EPR18352, 1:500, ab211926, Abcam, Cambridge, UK) or collagen α1 (I) (E8F4L, 1:100, #72026, Cell Signaling, Danvers, MA, USA) for 30 min at 32 °C. The reactions were detected using the standard polymer technique EnVision™ FLEX/HRP Detection Reagent (GV800, Dako Omnis, Agilent), and signal intensity was enhanced using the EnVision™ FLEX+ Rabbit LINKER (GV809, Dako Omnis, Agilent) and visualized using EnVision™ Flex DAB+ Chromogen system (GV825, Dako Omnis, Agilent). Finally, the sections were counterstained with hematoxylin and mounted with Pertex. The immunostained sections were evaluated by a senior consultant pathologist (L.M.R.G.).

### 4.7. Studies in NHEKs and NHDFs

NHEKs (C-12006) were derived from 24- to 57-year-old Caucasian women and purchased from PromoCell (Heidelberg, Germany). NHEKs were cultured in keratinocyte growth medium-2 medium (PromoCell) composed of keratinocyte basal medium (KBM)-2 with penicillin (100 IU/mL), streptomycin (100 μg/mL), and amphotericin-B (50 ng/mL) and supplemented with bovine pituitary extract (30 µg/mL), recombinant human epidermal growth factor (0.125 ng/mL), insulin (5 µg/mL) and transferrin (10 µg/mL), hydrocortisone (0.33 µg/mL), epinephrine (0.39 µg/mL) and CaCl_2_ (0.06 mM) and on collagen I-coated surfaces. NHDFs (CC-2511) were derived from a 37-year-old Caucasian woman and purchased from Lonza (Basel, Switzerland). NHDFs were cultured in DMEM with GlutaMAX™, glucose (4.5 g/L) and pyruvate (Gibco, Life Technologies, Grand Island, NY, USA) with 10% fetal bovine serum (FBS; Gibco heat-inactivated qualified FBS, 10500064, Thermo Fisher Scientific, Waltham, MA USA), penicillin (100 IU/mL) and streptomycin (100 μg/mL). Cells were incubated in a humidified atmosphere of 5% CO_2_/air at 37 °C and were passaged using 0.05% trypsin-0.02% EDTA (Biological Industries, Kibbutz Bet-Haemek, Israel) [24].

NHEKs and NHDFs were seeded (1 × 10^5^ in 1 mL/well) in 24-well tissue culture plates (CellStar^®^, Greiner Bio-One). NHEKs were grown in wells coated with collagen I [14]. Cells were incubated for 72 h. The confluent cell layers were then washed with Dulbecco’s PBS (pH 7.4) and starved for 24 h in serum-free KBM-2/DMEM with 1.8 mM CaCl_2_ containing 1 mg/mL bovine serum albumin. The cells were then treated with 30 ng/mL rhIL-6 (206-IL, R&D Systems, Minneapolis, MN, USA) and 1 ng/mL rhIL-1β (201-LB, R&D Systems) separately and combined for 24 h [62]. Media were collected, spun (2000× *g*, 10 min, 4 °C) and the supernatants kept at −80 °C until analysis for LDH activity [14], and MMP-1 (RAB0361, Sigma-Aldrich, St. Louis, MO, USA) and TIMP-1 (ab187394, Abcam) levels by ELISA [1,63]. 

Total RNA of the treated NHEKs and NHDFs was extracted with 1 mL of TriReagent^®^ (Molecular Research Center, Cincinnati, OH, USA). Bromochloropropane (100 µL) was added to isolate the aqueous phase containing the RNA, which was precipitated using isopropanol. The RNA pellet was then washed in ethanol and subsequently dissolved in 10 μL RNAse-free water. Total RNA concentrations were determined with the Ribo-Green assay (R11490, Life Technologies).

Total RNA (500 ng) was converted into cDNA in 20 μL using OmniScript reverse transcriptase (Qiagen, Valencia, CA, USA) and 1 μM poly-dT (Invitrogen) according to the manufacturer’s protocol (Qiagen). For each target mRNA, 0.5 μL of cDNA was amplified in 25 μL of SYBR Green polymerase chain reaction (PCR) containing 1× QuantiTect SYBR Green Master Mix (Qiagen) and 100 nM of each primer, as shown in Table 2. The amplification was monitored in real time using an MX3005P Real-time PCR machine (Stratagene, La Jolla, CA, USA). Ct values were related to a standard curve made with known concentrations of cloned PCR products or DNA oligonucleotides (Ultramer™ oligos, Integrated DNA Technologies, Leuven, Belgium) with a DNA sequence corresponding to the sequence of the expected PCR product. The specificity of the PCR products was confirmed by melting curve analysis after amplification. RPLP0 mRNA was chosen as an internal control.

### 4.8. Statistical Analysis 

DEGs were identified by the paired t test (participant eliminated as factor, *p* < 0.05, log2FC ≥ |1.00|). All data were log2-transformed prior to analysis using Qlucore Omics Explorer software, version 3.7 (Qlucore AB, Lund, Sweden). Gene expression correlations with *MMP1* and *MKI67* were calculated using Pearson’s correlation coefficients. Cell culture data (LDH, BrdU incorporation, RT–qPCR and ELISAs) were analyzed with one-way ANOVA and the Holm-Sidak post hoc method using SigmaPlot software, version 14.0 (Systat, Palo Alto, CA, USA). The level of statistical significance was set to *p* < 0.05. FDR-adjusted *p* values, i.e., *q* values, were calculated [64].

## 5. Conclusions

The major genes involved in human epidermal wound healing were successfully quantified using a customized panel for the NanoString platform. The obtained wound healing gene expression signature, consisting of 28 DEGs, is a start in identifying possible therapeutic targets to accelerate reepithelialization and epidermal stratification. The overlapping functions of upregulated cytokines/chemokines and growth factors could indicate biological redundancies.

## Figures and Tables

**Figure 1 ijms-23-15746-f001:**
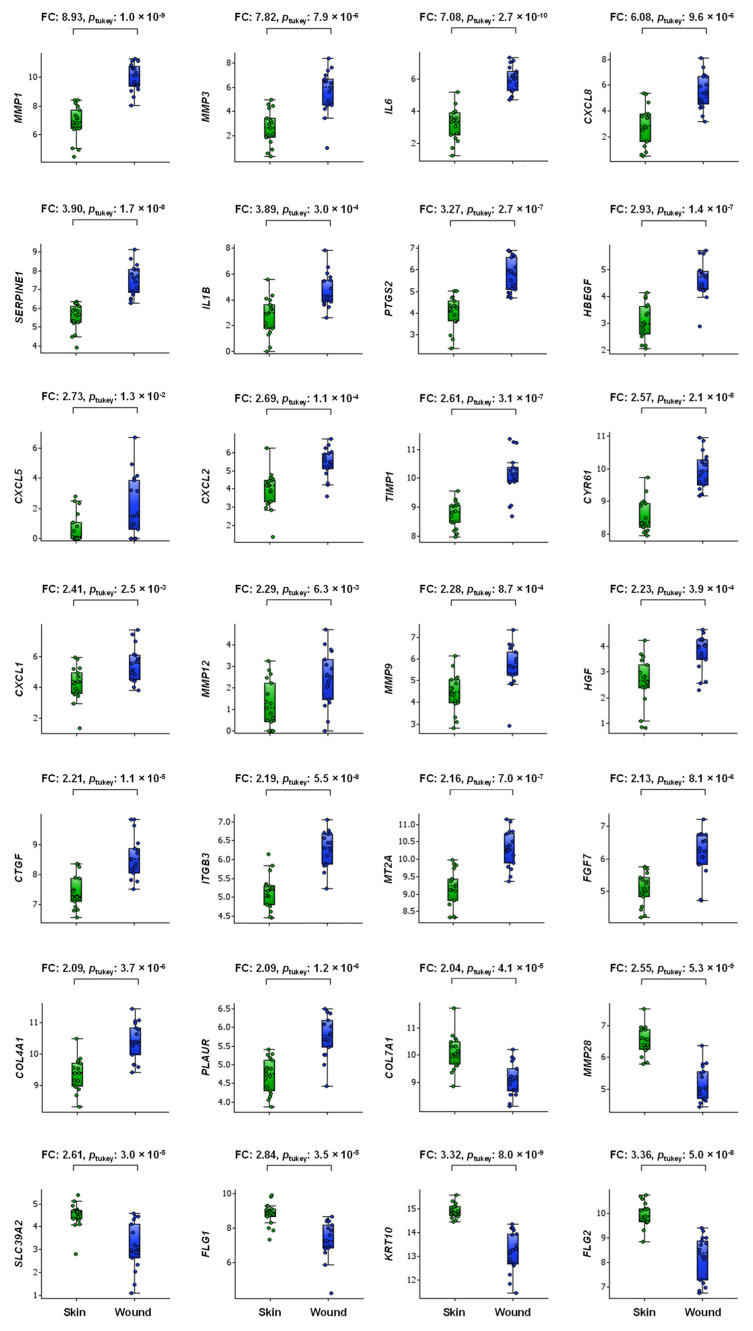
Box plot of injury-induced DEGs in wound tissue (blue symbols) vs. adjacent normal skin (green symbols) tissue of 4-day-old deroofed suction blisters (*n* = 17) arranged in descending FC order. Boxes represent the 25th–75th percentiles, whiskers represent the 5th–95th percentiles, and the horizontal dashed lines within the boxes indicate the median values. The y-axis shows log2-transformed expression values. FC, fold change.

**Figure 2 ijms-23-15746-f002:**
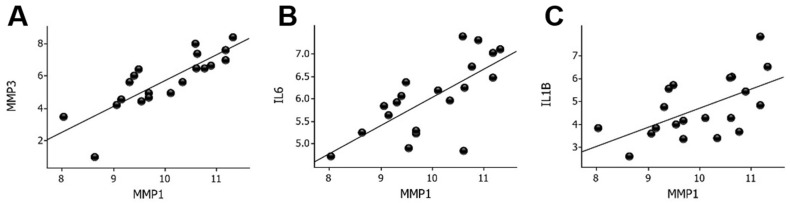
Correlation between *MMP1* mRNA levels and *MMP3* (**A**), *IL6* (**B**), and *IL1B* (**C**) mRNA levels in wounds. Log2-transformed expression values are shown on the y- and x-axes. MMP, matrix metalloproteinase; IL, interleukin.

**Figure 3 ijms-23-15746-f003:**
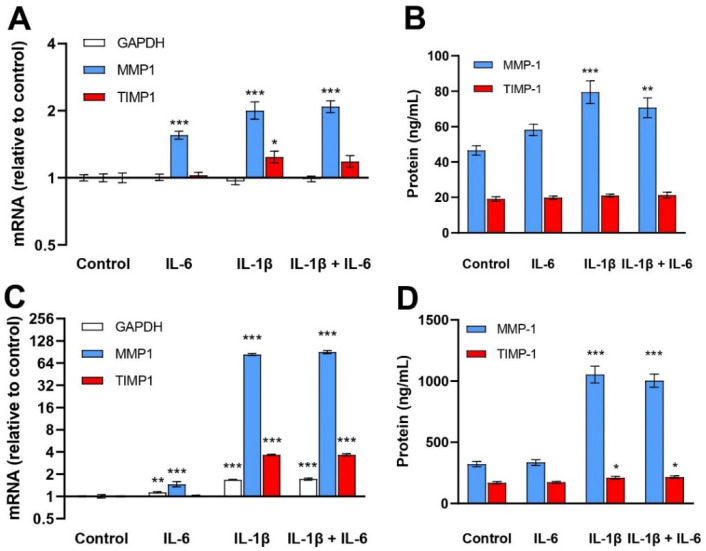
Effects of IL-6 (30 ng/mL) and IL-1β (1 ng/mL) treatment of NHEKs (**A**,**B**) and NHDFs (**C**,**D**) on *MMP1* and *TIMP1* mRNA (**A**,**C**) and protein media levels of MMP-1 and TIMP-1 (**B**,**D**). mRNA levels were determined by RT–qPCR and normalized to *RPLP0*, expressed as FC relative to control-treated NHEKs and NHDFs. The geometric mean ± back-transformed SEM of 6 replicates is shown (**A**,**C**). MMP-1 and TIMP-1 protein levels were determined by ELISA. The mean ± SEM of 6 replicates is shown (**B**,**D**). * *p* < 0.05, ** *p* < 0.01, *** *p* < 0.001 vs. control.

**Figure 4 ijms-23-15746-f004:**
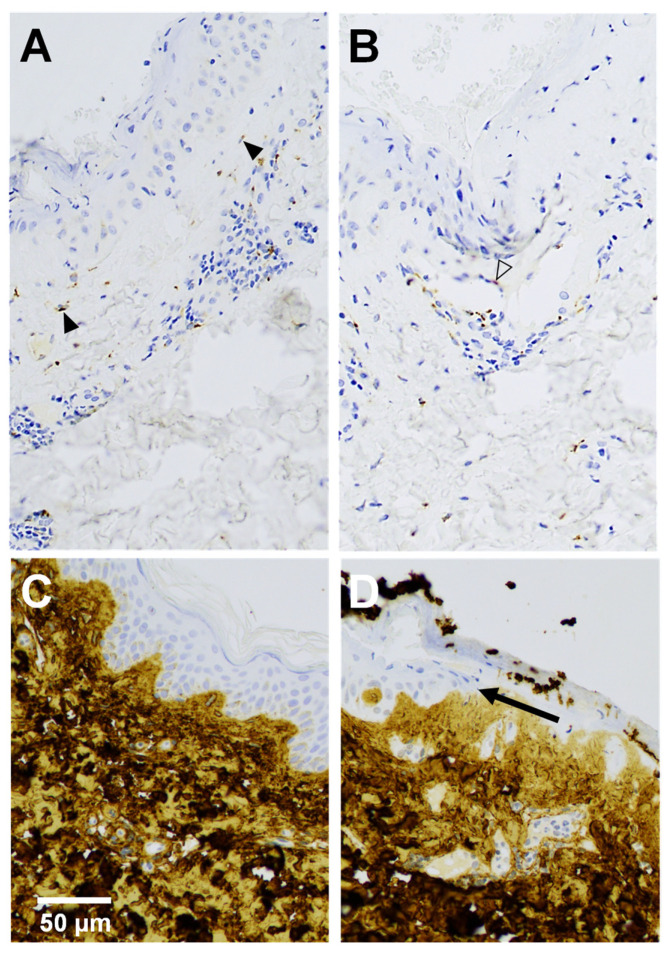
TIMP-1 (**A**,**B**) and collagen α1 (I) (**C**,**D**) immunohistochemical analysis of a 4-day-old wound (**A**,**B**,**D**) with adjacent skin (**C**). (**A**) Fibroblasts (solid arrowheads) below neoepidermis and (**B**) endothelial cell (outline arrowhead) of small blood vessel show positive immunoreactivity of TIMP-1 in upper dermis. (**D**) Arrow indicates the tip of the migrating neoepidermal tongue.

**Figure 5 ijms-23-15746-f005:**
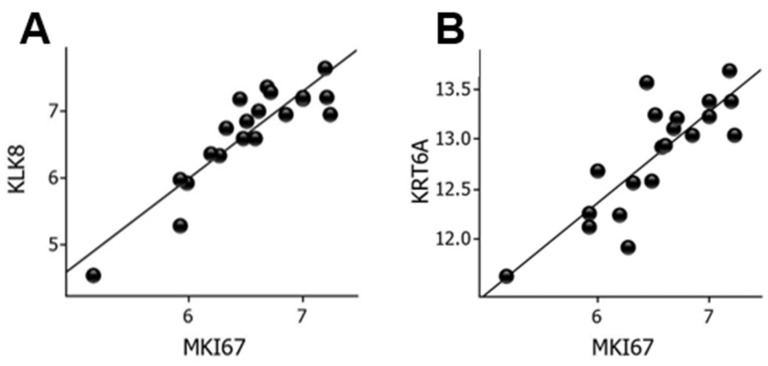
Correlation between *MKI67* and *KLK8* mRNA levels (**A**) and between *MKI67* and *KRT6A* mRNA levels (**B**) in wounds. Log2-transformed expression values are shown on the y- and x-axes. KLK, kallikrein; KRT, keratin.

**Figure 6 ijms-23-15746-f006:**
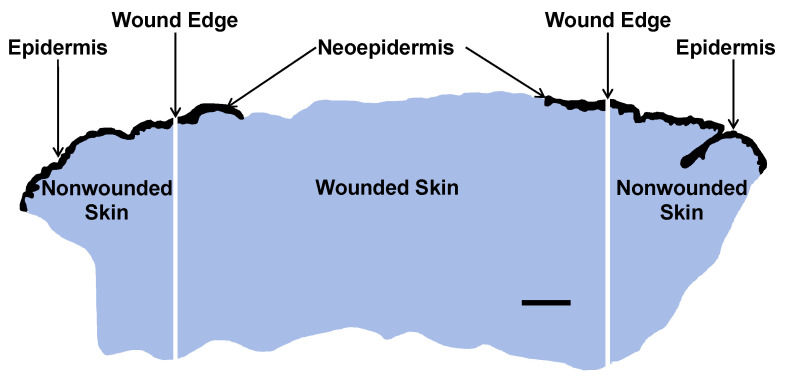
Macrodissection of FFPE tissue section of a day-4 suction blister wound into central wound (1 piece) including neoepidermis and adjoining nonwounded normal skin including epidermis (2 pieces). Reepithelialization was calculated using the following formula: Neoepidermis (mm)/Wound length (mm) × 100%. Total RNA was extracted, and the extracts were subjected to nCounter mRNA analysis. Epidermal compartments are black and dermal compartments blue. Scale bar, 1 mm.

**Table 1 ijms-23-15746-t001:** LDH activity (mU/mL) in media from treated NHEKs and NHDFs.

Cell Type	Control	IL-6	IL-1β	IL-1β + IL-6	*p* Value ^1^
NHEKs	25.9 ± 7.4	20.9 ± 6.3	19.1 ± 6.5	24.3 ± 8.4	0.370
NHDFs	14.7 ± 0.6	14.5 ± 0.9	13.7 ± 0.8	14.2 ± 1.5	0.377

^1^ One-way ANOVA. Mean ± SD of 6 replicates.

**Table 2 ijms-23-15746-t002:** Primer sequences for RT–qPCR analyses of treated NHEKs and NHDFs.

Gene	GenBank ID	Sense (Forward)	Antisense (Reverse)
*MMP1*	NM_002421.4	CGAATTTGCCGACAGAGATGAAG	GGGAAGCCAAAGGAGCTGTAGATG
*MMP3*	NM_002422.5	GATCCTGCTTTGTCCTTTGATGCTGT	CTGAGGGATTTGCGCCAAAAGTG
*TIMP1*	NM_003254.3	CGGGGCTTCACCAAGACCTACA	TGGTCCGTCCACAAGCAATGA
*GAPDH*	NM_002046.4	CCTCCTGCACCACCAACTGCTT	GAGGGGCCATCCACAGTCTTCT
*RPLP0*	NM_053275.3	GGAAACTCTGCATTCTCGCTTCCT	CCAGGACTCGTTTGTACCCGTTG

## Data Availability

The data presented in this study are available on request from the corresponding author.

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
