# Peer review of "Gene Expression Linked to Reepithelialization of Human Skin Wounds"

_ijms, 2022, doi:10.3390/ijms232415746_

Round 1

Reviewer 1 Report

The reaserch is very interesting and well done. I have few observations:

1-In the title please add .....reepithelization of  suction-blister wound in human skin

2- It is not correct to designate the areas of the studied specimen as skin, edge and wound. The wound is also skin. I suggest to call them as nonwounded skin, edge of wound and wounded skin. Precise terms are very important in science

Author Response

Thank you for reviewing our manuscript and for provided valuable comments.

Our responses to your review:

1.    We agree that including suction blister in the title would be technically more correct. However, to attract maximal number of readers we prefer to maintain the more general title but we now specify in the Abstract that we used the suction blister wound model. I hope this is satisfactory to you.
2.    Thank you for making us paying more attention to the details in Figure 6 and for having provided suggestions for improvements. We have tried to implement these in the redrawn Figure 6.

Reviewer 2 Report

Summary:

The aim of this study was to validate a customized gene expression panel using archival FFPE tissues of standardized epidermal wounds from a randomized controlled trial in healthy volunteers.The authors analyzed FFPE tissues from 10mm epidermal wounds induced by the suction blister technique, including adjacent normal skin from 20 individuals representing the control arm of a randomized, double-blind trial comprising 30 healthy volunteers.

Using Nanostring technology out of a customized panel of 139 target genes the authors identified 22 upregulated differentially expressed genes judged essential for reepithelialization and 6 downregulated differentially expressed genes determined to be associated with epidermal maturation.

These findings may help to identify therapeutic targets to accelerate reepithelialization and epidermal stratification

Critique:

Based on the detailed analysis of each of the genes identified, it appears that the goals of the study, ie, to validate the testing platform was accomplished. However, to add relevance for future wound healing studies, please address the following comments:

1)     Normal volunteers – 10 of the 30 healthy volunteers were not included.  Please explain in the methods section why these volunteers were not included and if they had different attributes from the 20 who were included.

2)     Although the genes identified are certainly involved in wound healing, it is not clear that they are all involved in epithelialization.  Specifically, TIMP1 was identified in multiple cellular locations and from this study it is not possible to discern whether these cells communicate with the keratinocytes to promote either proliferation or migration

3)     11 of the Preselected Human Wound Healing Panel and 11 of the ‘custom’ genes were differentially expressed. Please comment on the relevance of this finding given that the Preselected panel is supposed to represent genes important to wound healing.  Specifically could the final 28 DEGs be narrowed to a ‘minimun data set’ and if so, how would this be accomplished.

4)     The discussion is very long and complex but points out that most of the markers analyzed were already known. Recommend condensing the discussion and highlighting the most important findings of the study rather than a complex literature review.

5)     Normal volunteers were utilized but not compared to impaired epithelialization. Please comment on how the findings from this study could be used in patients with impaired wound healing.

Author Response

We are very grateful for the comments and constructive criticism.

  1. Thank you for pointing out this deficiency in our manuscript. The design of the trial and enrollment criteria of the healthy volunteers are described in more detail now (highlighted in yellow). The control arm of 20 participants did not differ from the entire study population of 30 participants although the distribution of sex differed somewhat (40% versus 50% women).
  2. The reviewer brings up a very relevant issue. Although TIMP-1 was expressed in dermis it may indirectly influence reepithelialization by a positive influence of angiogenesis. Also, the suction blister model does no leave a scar which indicates minimal involvement of dermis. This is now discussed on page 7, lines 237-239 (highlighted in yellow). "Neither COL1A1 expression nor proliferation of stromal cells were increased in dermis [1]. The lack of a fibroproliferative response explains the fact that these lesions leave no scar." Nevertheless, we have omitted the sentence in the Abstract "... judged essential for reepithelialization ..." Instead we added a sentence in the 1st paragraph of the Discussion: " ... and all of these genes are considered important for reepithelialization." 
  3. The Preselected NanoString Wound Healing panel was developed for all types of organs and tissues. We added genes more related to wound healing processes in the skin specifically. The reviewer is certainly correct in that the 28 DEGs may be narrowed down to a smaller, "minimum" gene set, as many of the genes correlate with each other as shown for MMP1 and MMP3 (please find the graph in attached PDF file). Thus, any of the two genes, MMP1 or MMP3, may replace each other. However, because we cannot biologically justify why one should be more important than the other, and because we would rather have a complete view of the wound profile (“fear of missing out” .-), we have decided to report the full set of DEGs, as these constitute the actual wound signature. We hope that the respected reviewer understands our decision.
  4. The Discussion has been shortened (by 20%) and more focused on the main findings.
  5. We appreciate very much that the reviewer brings this up. We have added our interpretations and extrapolation of the results in the last 2 sentences of the Abstract as follows (highlighted in yellow): "The observed gene expression changes correlate well with the current knowledge of physiological reepithelialization. Thus, the gene expression panel described in this paper could be used in patients with impaired healing to identify possible therapeutic targets."

Author Response

Thank you very much for the very relevant and useful suggestions.

Abstract

  • We have added the main impact of the study and the perspective in the last 2 sentences (highlighted in yellow).
  • 6 to six has been changed.

Introduction

  • We have supplemented the Introduction with a paragraph on page 2, lines 49-54 of the use of the suction blister model and the need for more molecular data (highlighted in yellow).

Results

  • A scale bar has been added to Figure 4.

Method

    • Thank you for reminding us of listing the inclusion and exclusion criteria. These are now found on page 9, lines 298-301.
    • The tissue material was fixed in formalin and embedded in paraffin to enable histology and immunohistochemical studies. It is also more difficult to macrodissect fresh tissue as opposed to formalin-fixed, paraffin-embedded tissue.
    • We have now included more details in Figure 6 with the hope this is now satisfactory .